# Establishing an Air Quality Index Based on Proxy Data for Urban Planning Part 1: Methodological Developments and Preliminary Tests

Claudia Falzone * and Anne-Claude Romain

Research Unit SPHERE, Department of Environmental Sciences and Management, University of Liege, 4000 Liège, Belgium
* Correspondence: cfalzone@uliege.be; Tel.: +32-63-230-849

**Abstract:** In the last few decades, urban planning has expanded regarding environmental considerations. However, air quality, which is regarded as an important aspect of the green development of cities, is not considered in urban planning. This research aims to propose a tool to easily introduce air quality considerations into urban projects. Nowadays, the usual air pollutants ($NO_x$, PM, $SO_2$, and $O_3$) are measured via sophisticated monitoring stations—or even low-cost devices—to give near-real-time air quality indices. However, stations are not adapted to local air pollution and real-time data are not helpful for planning purposes. An index able to rank areas and projects based on urban "air proxy data" would help decision makers. This paper presents how to create an air quality index as a decision support tool for urban planning. No pollutant measurement campaign will be necessary and only data that are easily accessible, even to nonexperts, are used. This paper describes the methodological development of an index that we call AQOI (Air Quality Observed Index), and the results obtained for four different locations (industrial, urban, and rural) considered as preliminary tests.

**Keywords:** air quality index; urban planning; air pollution; proxy data; local scale

## 1. Introduction

Across the world, urban areas of industrialized countries are affected by pollution, which is mostly linked to anthropogenic activities (traffic, domestic heating and activities, industries) and, to a lesser extent, to natural sources, such as vegetation, volcanoes, or forest fires. The main anthropogenic pollutants are continuously produced within cities, and their dispersion is essentially prevented by the urban architecture. Pollutant concentration measurements are different in each city due to discrepancies in the control of pollution sources, meteorology, and human behavior [1]. The concentration levels of pollutants are primarily influenced by the quantities emitted into the ambient air, to which are added the local meteorological conditions that govern their dispersion, conversion, and deposition (wet or dry) [1]. A pollutant can be of primary or secondary type, depending on whether it is emitted directly by the source (e.g., $NO_2$ from traffic) or the result of conversion phenomena (e.g., ozone, which is generated when nitrogen oxides ($NO_x$) combine in the presence of sunlight). Moreover, certain pollutants, such as particle matter (PM), have different compositions depending on the source (e.g., domestic fuel, wood burning) [2].

In 2016, the International Agency for Research on Cancer (IARC) classified outdoor air pollution as a carcinogenic agent in humans (vol. 109) [3]. To keep an eye on the matter, air pollutants are continuously monitored by fixed networks present in each European country. Concentrations must comply with the limit values described in the 2008 European directive [4]. In the coming years, the threshold values set out in the European directive will be reviewed on the basis of new guidance values issued by the World Health Organization [5,6]. In its 2020 report, the European Environment Agency (EEA) showed a decrease

in $NO_2$, $PM_{10}$, and $PM_{2.5}$ for each type of station (urban, suburban, rural, industrial, and traffic), which is reflected in the number of years of life lost (YLL) (in 41 European countries): 4,806,000 and 624,000 in 2018 for $PM_{2.5}$ and $NO_2$, respectively, compared to 5,723,600 and 1,414,100 YLL in 2009 for $PM_{2.5}$ and $NO_2$, respectively [7].

To communicate with citizens about air quality, indices are created based on the concentrations monitored by network air quality stations. The aim of the research presented herein is the establishment of a global index called an Air Quality Observed Index (AQOI) that reflects the level of air quality. It is based solely on open-source data from one year. The AQOI is designed in such a way that it is easy to use, inexpensive, and relevant for various simulations. Finally, this index is intended to be a useful decision-making tool based on air quality for the implantation of new urban projects.

After the selection of variables involved in the AQOI calculation, tests were conducted on a selected study site to report and comment on the new index's performance.

## 2. Air Quality Monitoring and Urban Management

### 2.1. Monitoring

Certain pollutant concentrations are given instantly by on-site analyzers; others are collected by sampling devices and must be analyzed in a laboratory (e.g., the gravimetric method for the PM measurement). Analyzers must pass a certification test before being accepted into the network. For example, the PM measurement by an analyzer must be subject to the equivalence method (in comparison with the gravimetric method) [8]. A fixed network consists of reference devices that are subject to regular maintenance. The analytical techniques of analyzers make them state-of-the-art instruments capable of taking instant and accurate measurements at low concentrations. Consequently, they are very expensive.

Recent years have seen the emergence of low-cost instruments sensitive to low concentrations on the order of ppbv (part per billion volume) following the development of sensors specific to certain pollutants (e.g., electrochemical sensors). However, these instruments have lower performance than analyzers [9,10] and are not yet subject to regulations. This is, however, a work in progress [11]. The document relating to the performance evaluation of air quality sensor systems (Part 1: Gaseous pollutants in ambient air) has been available since 17 December 2021 (CEN/TC 264—17660-1). In view of the rapid development of this type of instrumentation, we will soon see its implementation in the network as a support to fixed networks, allowing for the monitoring of local pollution in real time.

In addition to the measuring instruments on the field, other tools have been developed, such as spatial interpolation and dispersion models. Spatial interpolation consists of employing the land-use regression model (LUR) that applies to the urban scale. This requires information from a Geographic Information System (GIS) (e.g., road traffic, land use, altitude) and the concentrations of pollutants measured at monitoring sites. With this, it creates a map with extrapolated concentrations in unmonitored areas [12,13]. Dispersion models are numerous and involve various methods of calculation. The calculation model differs according to the scale at which the study is to be carried out. At the micro-scale, Computational Fluid Dynamics (CFD) are considered. Modelling allows different scenarios to be predicted for chosen daily periods depending on the weather conditions, architectural configuration, and variability of sources. However, this method is often time-consuming, costly, and difficult to implement within practical turn-around periods [14].

To facilitate communication with citizens and assess the level of pollution, indices have been developed. They are based on a two-step calculation: (1) for each target pollutant, categories (good to poor) are defined based on chemical concentration ranges; (2) the pollutant with the worst (poorest) category defines the level of the index. Table S1 shows the U.S., and EU Air Quality Index (AQI), each with six categories, and the Belgian Air Quality Index (BelAQI) with 10 categories [15,16]. The BelAQI is based on a concept developed in the European JOAQUIN project [17]. The results of the HRAPIE project enabled the JAOQUIN project to develop different categories with the help of the document "Health risks of air pollution in Europe" (HRAPIE project) [18]. Although these indices are

based on pollutant levels, they provide information in a simple format reflecting a complex situation, which makes them a very handy tool for decision-making. The main drawback is that they do not provide values far from the analyzers and the estimation by models is inaccurate in specific areas, such as streets with a canyon configuration. Moreover, it is to be noted that the comparison of air quality between small areas and between sites is not possible based on indices alone, as the index value is determined by the worst pollutant, which may differ from site to site [19].

### 2.2. Urban Management

The authorities implement strategies to positively influence urban air quality by managing traffic, integrating green infrastructures, and lowering the energy consumption of buildings. The currently available tools for such actions are not suitable at a local scale (e.g., AQI) or are too expensive (e.g., analyzers) and time-consuming (e.g., modeling). Low-cost instruments allow for real-time monitoring but do not give any prediction of the air quality status if the site configuration changes. Moreover, the multiplication of these instruments would represent a non-negligible cost for a city. However, it is possible to make predictions if these tools are used together. The combined use of low-cost sensors and/or an official network with pollutant dispersion modeling can provide these predictions by modifying the urban architecture of the modeled area. It is also important to mention that the use of dispersion models and low-cost devices requires a degree of data interpretation that involves knowledge of the field, which is not the case in the AQI, and a high level of expertise. It will be interesting to develop an index that allows authorities to easily study local air quality for current and future planning configurations without having specific air quality expertise. Moreover, its use would be facilitated through a GIS query allowing for visualization of the index on a map. This index would allow the authorities to have a general view of the state of air quality, and therefore, take responsible urban planning actions accordingly.

## 3. Methodology

### 3.1. Conceptualization of AQOI

A global index called the Air Quality Observed Index (AQOI) that reflects the level of air quality is developed. It is based solely on open-source data taken over one year. The development of the AQOI was performed in three steps (Figure 1). The first step was to identify, in the literature, the elements considered as sources or sinks, as well as the dispersion phenomena. Based on these different elements, variables with varying complexity levels have been identified. At the same time, an experimental site was chosen to develop the indicator with experimental data (see below). The second step was to impose a limit area for the AQOI calculation and search the internet for free proxy data containing as many of the variables identified in the first step as possible. Then, the data following the limit area were extracted. The final step was the development of the algorithms in Access (Microsoft Office®, Microsoft Corporation, Redmond, WA, USA) to obtain the final AQOI value. Thus, the data extracted from the GIS were stored in the Access tables and the algorithms were applied through "queries" (Access terminology). In parallel, the process started from the second step, and was reproduced in a GIS (data extraction and application of the "queries"). This paper presents the variables used for the calculation of the AQOI, the limit area, and the algorithms to obtain the Air Quality Observed Index.

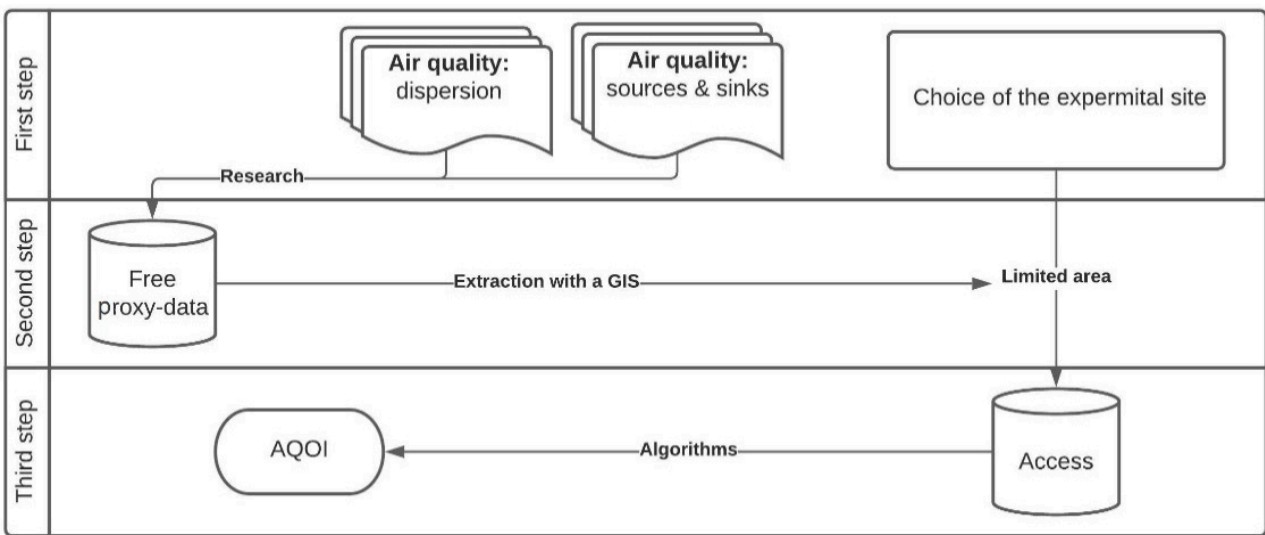

**Figure 1.** Flow scheme of the AQOI conceptualization.

The experimental site to determine the feasibility of the AQOI concept was "the administrative city of Seraing", which is a specific building in Seraing, located in eastern Belgium (50°35′ N; 5°30′ E) (Figure 2). The city covers a 35.34 km² area, is crossed by the Meuse, and lies at an altitude of 61 m a.s.l. The climate is temperate–oceanic, and the annual average temperature is 9.8 °C, with an average in winter of 2.4 °C and 17.7 °C in summer. The average annual rainfall is 772 mm [20]. The population density is 1800 inhabitants/km² [21].

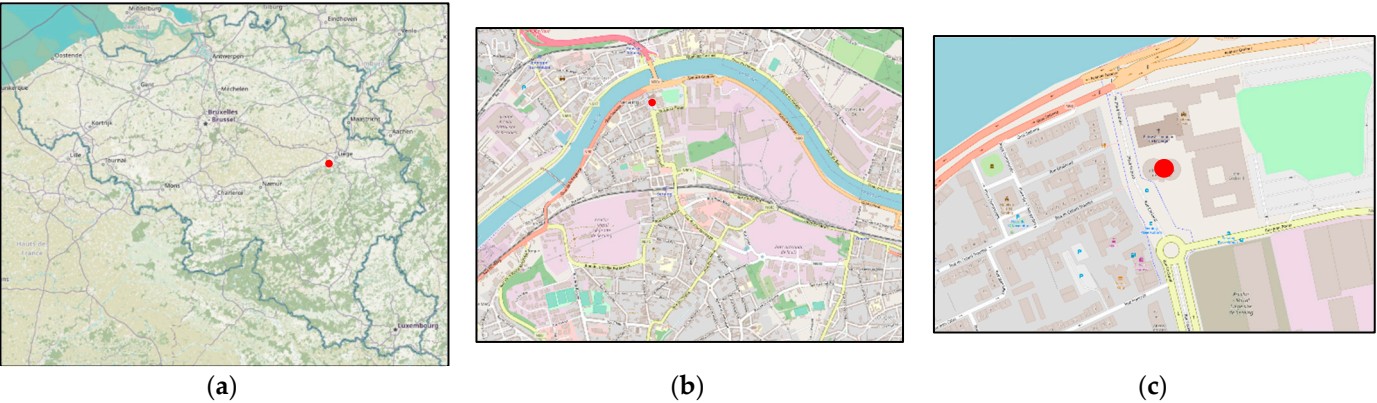

(**a**)          (**b**)          (**c**)

**Figure 2.** Location of the experimental site: (**a**) Map of Belgium with Seraing in red; (**b**) map of Seraing; (**c**) Seraing administrative city. Map data © OpenStreetMap contributors, CC BY-SA 2.0.

### 3.2. Modus Operandi of AQOI

Although developed in Access, the index has been designed to be calculated directly in a GIS. The procedure for performing the AQOI calculation is shown in Figure S1 (the flow scheme summarizing the process of the AQOI calculation via a GIS). The operator starts by choosing the study site in a GIS. By default, a limited area is created around the site, as well as a grid in which the index will be calculated. The proxy data are then extracted from the Web Map Service (WMS) layer or a data table with georeferencing from specific websites, but sometimes manual input by the operator is necessary. Based on the data, a routine is executed to obtain the AQOI value, which is then converted into a qualitative variable ("color") and represented in the GIS via a heat map.

## 4. Definition of the Area of Interest

The cell size selected for the AQOI calculation has been chosen according to the different values encountered in the LUR models [12,13,22]. To facilitate the extraction and interpretation of the data, the cell size has been defined as a square area with 300-m sides, called an "Index Cell" (IC), which has been divided into nine subcells (s-IC) (Figure 3a).

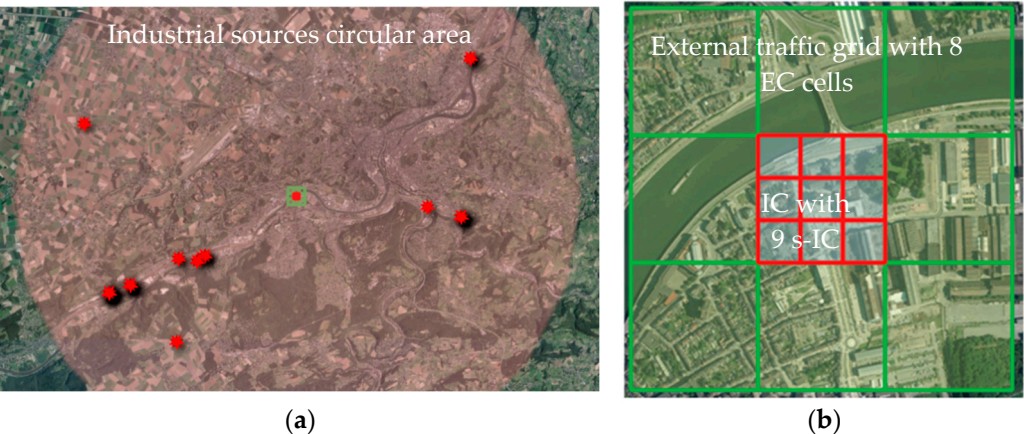

**(a)**                  **(b)**

**Figure 3.** Areas containing variables useful for the calculation of the AQOI: (**a**) Circular area considered for the location of industrial sources of pollution potentially influencing IC air quality, in pink (radius: 15 km). Red spots represent industrial sources. (**b**) Index cell IC (300 m × 300 m) (in red) with its nine subcells (s-IC). IC is centered on the administration building of Seraing. External cells (EC) in green (Service Public de Wallonie, Orthophotos 2018—Service de visualization REST. Visualization with QGIS V3.16.7).

Two factors, representing external sources able to influence the air quality into the IC, were considered. The first was the road traffic near the IC; therefore, a grid called "External Cells" (EC—Figure 3a) was created around the IC itself. The second concerned the industrial sources (IED—Industries subject to the European Directive) within a 15 km radius around the IC (Figure 3b). In the case of a Gaussian dispersion of the plume of pollution emitted by the industries, this distance includes almost all the parameters $\sigma_y$ and $\sigma_z$ according to the various classes of Pasquil stabilities [23].

## 5. Structure of the Variables and Parameters Involved within the Index

Four key elements of the urban area that can influence the air quality were selected for the IC, namely Topography, Buildings, Roads, and Vegetation. Since the IC characteristics are influenced by the neighboring cells and eventually by the industrial pollution plume from local industries, they were named "External Sources". Indeed, some of the variables defining these parameters can have an impact on IC air quality due to their influence on the dispersion of pollutants (such as the layout of buildings) or because they are a real source of pollution, such as traffic. Some have also been considered as sinks, such as vegetation favoring the dry deposition process [1].

Variables were selected based on their relevance to air quality, their accessibility at low cost (or free), and their ease of use. This led to us identifying 31 indicative variables (Figure 4). Their intrinsic value was obtained from observations and measurements, from data provided by administrations, and from an open-source database.

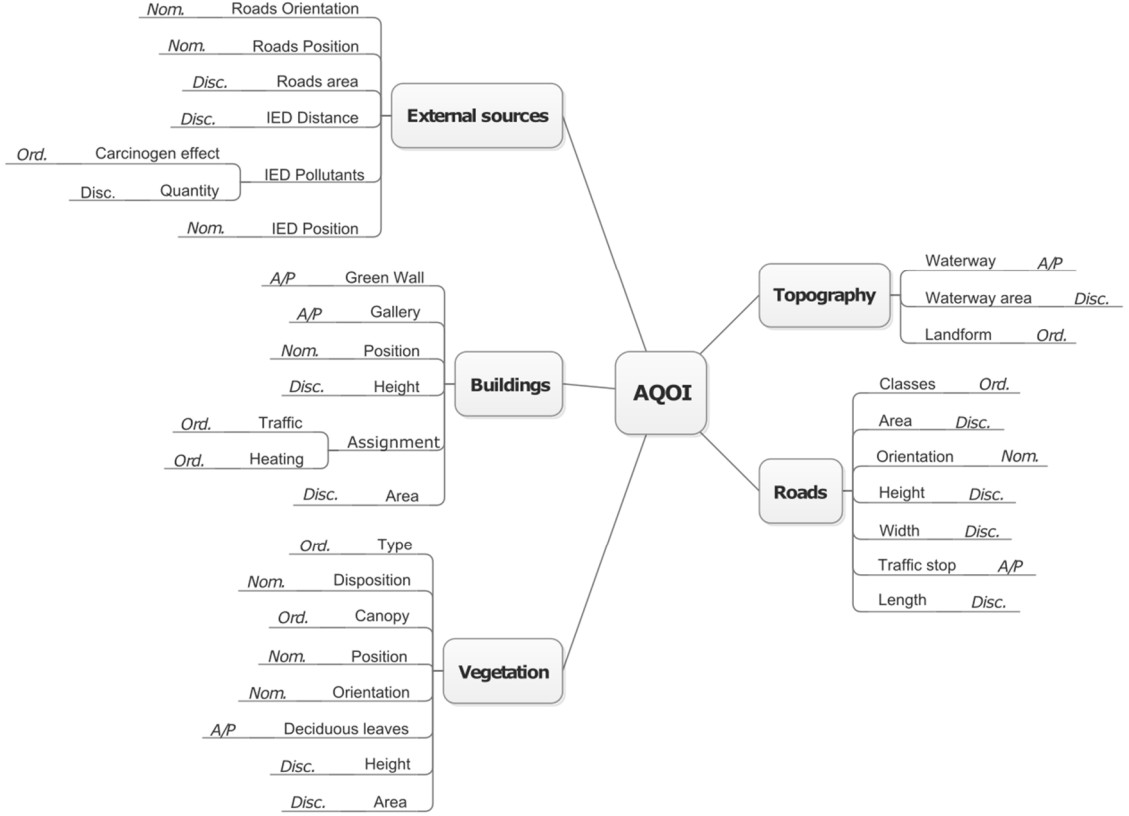

**Figure 4.** Parameters and their variables allowing for the calculation of the AQOI. The names of the variables and their type are presented (Disc. = discrete; A/P = absence/presence; Ord. = ordinal; Nom. = nominal).

There are four types of variables, obtained in the following way (Figure 4):

1. Discrete variables (*Disc.*): field measurements or extracted from GIS;
2. Absence/presence variables (*A/P*): value gathered by observation or extracted from GIS;
3. Ordinal variables (*Ord.*): extracted from GIS;
4. Nominal variables (*Nom.*): extracted directly or by observation into GIS.

A positive value is attributed to a variable whose impact on air quality is negative and vice versa. The neutral situation is expressed by zero. This selection was made based on the existing AQI with high values for poor air quality.

*5.1. Parameter Choice*

5.1.1. Topography

The term "topography" represents "the configuration of a surface including its relief and the position of its natural and man-made features" [24]. Among these different features, vegetation and buildings will be treated separately. Therefore, this parameter contains landform and waterway variables. Landforms influence the wind field and thus the dispersion of pollutants. Waterways allow barges to cross the land and constitute river traffic. The input of pollutants is significant in some areas where international barge traffic is allowed according to the European Pollutant Release and Transfer Register (E-PRTR: https://prtr.eea.europa.eu/#/diffemissionsair, accessed on 5 October 2021).

5.1.2. Buildings

Concerning air quality, the "Building" parameter can be considered from four viewpoints:

- Morphology, which impacts the dispersion of pollutants. Indeed, the layout of buildings directly affects the wind fields by creating eddies and, thus, trapping pollutants [25]. Features of buildings, such as height, roof shape, and the presence of

gallery (or porosity) influence the wind flow [26–29]. Most modeling programs represent buildings as rectangular boxes that may contain galleries, but the shape of the roof is rarely represented. The AQOI developed herein includes galleries and geometric variables.

- Building occupation is a "source" of pollution, in particular due to combustion processes: particulate matter (PM) emissions, the release of polycyclic aromatic hydrocarbons (PAHs), BTEX, nitrogen oxide ($NO_x$), organic carbon particles (OC), and black carbon (BC). The pollutants emitted vary according, for instance, to the type of fuel used [2].
- The typology of the building can directly influence the traffic and bring more or less traffic into the cell despite the typology of the road (a hospital, because of its frequent use, brings more traffic than a house).
- The presence of green walls on buildings contributes to decreasing local pollution. Gaseous pollutants are removed by absorption through leaf stomata or by adsorption on plant surfaces by dry deposition [30,31]. Nevertheless, the plant and tree species have a different impact on dry deposition due to leaf morphology and wax content [32].

### 5.1.3. Roads

Roads and traffic are strongly linked to air quality since street geometry impacts the dispersion of pollutants and traffic is the largest contributor to urban $NO_x$, PM, metal, and volatile organic compounds (VOCs) pollutant emissions [1].

- The geometry of a street influences a wind field on the ground and a phenomenon similar to a canyon can be produced in a street. The pollutants can be trapped due to the street canyon effect [33]. Street canyons are characterized by their aspect ratio, which has been defined as the height (H) of the canyon divided by its width (W): regular (H/W = 1), avenue (H/W < 0.5), and deep (H/W ≥ 2). In addition, the length (L) of the street can be considered as the road distance between two major intersections. Three categories of length are considered: short (L/H ≈ 3), medium (L/H ≈ 5), and long (L/H ≈ 7). The canyon can be considered symmetrical or not according to whether the heights on both sides are the same or not [34]. The higher concentration is situated on the leeward side. However, the concentration value also depends on other factors (e.g., the wind speed, the angle between the streets, the wind direction above the roof) [35].
- Concerning traffic, the main factors influencing pollution are the density and pollutant emissions (depending on the category of vehicle, acceleration, age of vehicle, etc.) [36–39]. However, since the evaluation of traffic density requires laborious counting campaigns, a compromise has been reached by using road assignments and the presence or absence of stops at street junction(s).

### 5.1.4. Vegetation

In the AQOI, the vegetation is represented by trees, hedges, grasses, and crops. The height of vegetation in the urban area can influence the wind flow and its density directly impacts the dispersion: an isolated tree or well-spaced trees do not block the wind flow, as the induced turbulence favors the dispersion of pollutants. On the other hand, a high density of trees or hedges blocks wind flow and is considered a barrier [40]. Moreover, trees or hedges in a street can lead to a canyon effect, decreasing air quality [41,42]. The seasonal effect, with leaf-on and leaf-off periods, impacts the horizontal and vertical dispersion [43]. The deposition depends on the velocity particle adsorption, the pollutant concentration, and the LAI (leaf area index) [41], as well as the plant species [44–46].

Besides their positive effects on pollution, plants and trees are also considered as a source of:

- Volatile organic compounds (VOCs) emissions, responsible for the formation of ozone and a fraction of PM [40,47,48];
- Pollen release (more or less allergenic depending on the species [49,50]);

- Indirectly, via ammonia emissions during fertilizer spreading on crops, PM formation [51];
- Pollutants emitted by equipment during vegetation maintenance [52].

According to its albedo and evapotranspiration, the vegetation contributes to decreasing the ambient temperature in an urban area and reduces the building's energy use for climate control [48,53].

Of all the variables listed, some are easily accessible visually on the site or on a map such as "height", "deciduous tree", "vegetation disposition", and "vegetation position".

### 5.1.5. External Sources

Traffic and IED industries are determined within the external cells (EC) adjacent to the IC. The potential entry of pollutants into the IC is considered; hence, only the position and orientation of the EC roads compared to the IC are meaningful. According to the wind direction and the position and orientation of external roads, pollutants can enter into the IC (the example of pollutants entering the IC: a north–south street located in the EC-north with respect to the IC and the wind comes from the north).

The impact of IED industries on the IC depends on the geographic situation and distance from the IC. Only IED industries are included in the AQOI because they are subject to a European directive [54], obliging them to report on their pollutant emissions. This information is freely accessible in open source on the website of the European Environment Agency. Among the 59 pollutants listed: 13 belong to category 1, 4 belong to category 2A, 10 belong to category 2b, 5 belong to category 3 and 27 are not classifiable or not included in the IARC list [3].

### 5.2. Wind Direction

Since AQOI is a static indicator and not a near-real-time index, atmosphere stability is not useful. The wind directions only are considered. There is no need to use other types of meteorological variables (such as sunshine).

The yearly frequency of each wind direction is calculated to meet the target: a state index based on one specific year or an annual mean of several years. The data come from weather stations or simulations. It is recommended that these variables are measured at a height of 10 m [55].

The wind direction is divided into 12 classes, $j$ of 30 degrees of arc ((15°–45°); (45°–75°), etc.). The frequencies ($f$) of the wind directions encountered over a year are used to weight the different variables. Some variables presented in Section 5.3 interact with the 12 wind directions and their frequencies to obtain different situations that may improve, degrade, or have no impact on air quality.

### 5.3. Variables and Description

The identified variables are shown in Figure 4. They have different categories of values that are presented in this chapter.

### 5.3.1. "Absence–Presence"

"Waterway" variable: It was decided to define the boat traffic only based on the Absence or Presence of a waterway in the IC. The main reason is that the E-PRTR values are not usable due to a lack of data refreshment and the scale grid is not adapted for the IC (E-PRTR grid: 5 km × 5 km). If a waterway is present in the IC, that means that pollutants can also come from the riverboat traffic and negatively impact the air quality of the IC (Absence = 0; Presence = +1).

"Gallery" variable: The presence of a gallery in a building included within the IC. It reflects the potential capacity of the building to trap the air and then slow down the air exchange. Otherwise, there is no impact (Absence = 0; Presence = +1).

"Green wall" variable: In general, and independently of the species used, the presence of a green wall on an IC building improves the air quality locally. The positive impact is

represented by a negative value. Without a green wall, there is no impact (Absence = 0; Presence = −1).

"Traffic stops" variable: Their presence on the road negatively affects the air quality (Absence = 0; Presence = +1; since there are currently more combustion engines than electric motors in the car fleet). The increase in pollutant emissions from the vehicles into the IC is considered due to several elements: the duration, the number of stopped vehicles with their engine running ("start and stop" technology is not considered for two reasons: (1) it does not work all the time, and (2) not all vehicles have this technology), and the reacceleration of the vehicle from the stop.

"Deciduous leaves" variable: In this case, the term "presence" represents a deciduous tree, such as an oak, included in the IC. The term "absence" represents an evergreen, such as a conifer. The presence of leaves (or needles) impacts air quality by dry deposition of pollutants [56,57]. In one year, the quantity of pollutants held by a tree is due to the persistence of its leaves. A deciduous tree is naked (i.e., leaf-off) for approximatively two seasons (autumn–winter) and its capacity to retain pollutants can be divided by 2. So, when the qualitative value is "presence", the impact is positive; otherwise, the impact is divided by 2 (Absence = +0.5; Presence = +1).

### 5.3.2. "Ordinal"

"Landform" is defined according to four levels: "hilltop", "plateau", "flank", and "valley floor". Its value is attributed according to the landform influence on pollutants dispersion and, consequently, on air quality. For calculation, the qualitative ordinal variable has been transformed into a quantitative one according to the impact of each situation on the dispersion of pollutants. "Valley floor" represents the worst case (the zone with likely pollutant accumulation; value equal to +1). "Hilltop" is the best case (value equal to −1) and "plateau" is considered neutral (0). "Flanks" have an impact that varies according to the canyon effect and is weighted by the wind direction as well as by the frequency of the wind blowing in each direction in one year (−1 to +1).

The "Assignment" variable is defined according to the nature of the description of the buildings from an administrative database (in Belgium, the "*Projet Informatique de Cartographie Continue*" (PICC)). The 19 descriptors are presented in Table S2 [58]. This variable can be associated with two types of sources: traffic-related and domestic heating-related pollution. The values of the "Affectation_Traffic" indicate the importance of the level on the traffic flow in the IC. The descriptors "station", "hospital", "industry", and "gas station" represent the worst cases (for example, one hospital induces higher traffic than one dwelling and a value of +1 is attributed). "Annex", "castle", and "water tower" are related to a less dense traffic (value equal to 0). The values for the others ("community house", etc.) are determined following the average of the attendance rate (the attendance rate is a function of the number of people and the descriptor under consideration, with certain particularities, such as the opening days for shops or public establishments, the number of residents and users for private buildings, etc.). The values of the "Affectation_Heating" are calculated based on the average consumption (in kWh/m$^2$) according to the typology of the building and the type of fuel used. The heating impact is calculated based on the heated area for each building.

The "Class" variable is defined according to the nature of the description of the roads from the PICC (the same description as in Open Street Map [59]). For nine descriptors, their impact on air quality is related to traffic density: the descriptor "highway" represents the worst case (value equal to +1), while a pedestrian path (soft or electric mobility with no direct pollution) is the best case (value equal to 0). For the other levels ("low traffic"—or "living street", as characterized by Open Street Map (OSM) –, "residential road", "tertiary road", "service road", "secondary road", "primary road"), the value is adjusted following the density of traffic attributed to the road.

The "Type" variable is defined according to three large categories of vegetation: "wooded", "herbaceous", and "crop". "Crop" represents the worst case (value equal to

+1) and "wooded" the best (value equal to −1). Indeed, several works have demonstrated that the "wooded" type has a filter-like effect on its environment [48,56,60] and that the "herbaceous" type has no real effect on air quality (value equal to 0). Harrison explains that the "crop" type participates in the formation of particulate matter [61]. The final value depends on the vegetation footprint relative to the surface of the IC.

The "Canopy" variable includes both types of canopies observed: narrow and wide. Wide is the worst case (value equal to +1) and narrow the best (value equal to 0). The canopy influences the horizontal dispersion, as Salmond demonstrates [43]. Indeed, when the canopy is wide, the pollutants are trapped below the tree during the leaf-on period. On the contrary, when the canopy is narrow, the pollutants are dispersed in the atmosphere.

The "IED pollutants" variable includes the various pollutants emitted into the air, listed in the Industrial Emission Directive (IED). Their emissions are recorded in the *European Pollutant Release and Transfer Register* (E-PRTR). This variable is linked to the carcinogenic effect, which is considered as an ordinal value. For each pollutant, the carcinogenicity is determined by the "overall evaluation of carcinogenicity to humans" database from the IARC (International Agency for Research on Cancer [3]). The pollutants belonging to group 1 ("carcinogenic to humans") have the worst note (value equal to +10) and those from group 4 ("probably not carcinogenic to humans") have the best note (value equal to 0).

5.3.3. "Discrete"

The discrete variables are directly measured, have a metric unit, and are used as is or categorized into classes. The classification allows for the attribution of an impact factor that will minimize, or not, an ordinal and/or an absence or presence variable.

"Height" (h) of building and vegetation. This influences the air quality via the screen effect. This variable is extracted from a GIS as a discrete variable but, in the AQOI model, it is transformed into an ordinal variable. The height is categorized into four classes ($\leq$5 m; 5–10 m; 10–15 m; >15 m), and each class is associated with a factor from 1 to 4, respectively.

"Height of the sides of the street" (H) is based on the heights (h) of buildings or vegetation on both sides of the road. The comparison between the heights on both sides of a road allows us to define the symmetry of the street and, thus, to calculate the H variable (if $h_{lower}/h_{bigger} \geq 0.55$ then $H = (h_{lower} + h_{bigger})/2$ else $H = (h_{lower} + h_{bigger})/4$).

"Width" of the road influences the air quality following the ratio between the height and the width of the road. This variable is extracted from a GIS.

"Length" of the road influences the air quality following the ratio between the length and the height of the road. This variable is extracted from a GIS.

The "Area" variables, from waterways, buildings, roads, vegetation, and external roads, are calculated directly with a GIS. Their use is essential to the weighting of the different elements, allowing for the calculation of a subindex integrating the final calculation (see Section 5.5).

The variable "IED Pollutants" is related to the quantities of pollutants emitted and not the carcinogenic effect. Therefore, this variable ("Quantity"), expressed in discrete form, reflects the pollution emitted by the IED industries.

"IED distance" represents the distance between the industry and the IC. This value is expressed in kilometers and cannot exceed 15 km based on the circular area of selection of IED industries.

5.3.4. "Nominal"

All variables described here need to be combined with other variables, in matrix form, to estimate the impact of a situation on air quality in the IC.

The "position" of the element (building or vegetation) with regard to the building project subcell is essential for determining the screen effect (Figure S2). The values are expressed as cardinal points. The same values are attributed to the external road position; in this case, the values do not relate to the sub-IC but, rather, to the external traffic grid (Figure 4).

"Orientation" from road, vegetation, and external road has four levels: "N–S", "W–E", "NW–SE", and "NE–SW". This variable directly affects the dispersion of pollutants, either by a simple canyon effect in the case of roads, or by a simple screen effect in the case of vegetation, or by the possibility for pollutants to enter the IC in the case of an external road.

The "IED Position" corresponds to the 12 sectors of the wind (30° of arc each). In some cases, the pollutants emitted by the industry can penetrate the IC.

"Disposition" from vegetation is only used when "type" equals "wooded". For instance, when the trees are planted linearly or clustered, they can be considered as a screen.

*5.4. Combinations of Variables*

Into the model AQOI, some variables must be combined to obtain a unique value:

- The height (H) of the road can be associated with the width (W) and the length (L) of the road to determine the type of canyon effect and canyon length. According to Vardoulakis, the H/W ratio influences the canyon [34]. Currently, H/W is categorized into three classes (H/W < 1; 1–2, and ≥2), and each class is associated with a factor from 1 to 3, respectively. Furthermore, the L/H ratio allows us to determine the type of canyon length. L/H is classified into three classes (H/W < 4; 4–6.5, and ≥6.5); the first class is assigned a factor of 0.5, the second class a factor of 1, and the third class a factor of 3.
- The position (building or vegetation) combined with the wind direction provides a value, which, depending on the situation, improves, degrades, or has no impact on air quality in the IC (e.g., N wind direction and N position = improvement = −1; N wind direction and S position = degradation = +1).
- "Orientation" (road) combined with the wind direction provides a value, which, following the canyon situation, improves, degrades, or has no impact on air quality in the IC (e.g., N wind direction and N–S orientation = improvement = 0; E wind direction and N–S orientation = degradation = +1).
- When the vegetation is linear, it is requested to introduce the orientation of the plantation. This orientation, combined with the wind direction, provides a value (e.g., N wind direction and N–S orientation = 0; E wind direction and N–S orientation = +1). This value is used as a factor that moderates another value (see Equation (12)).
- "Height", "disposition", and "deciduous leaves" of the vegetation, combined, provides a value, which, depending on the situation, improves, degrades, or has no impact on air quality in the IC (e.g., <5 m and presence of deciduous leaves and linear = 1; >15 m and absence of deciduous leaves and linear = 4). This value is used as a factor (see Equation (12)).
- The combination of "External road Position" and "External road Orientation" indicates if the external air pollutants from traffic penetrate the IC or not (e.g., NS orientation and N position = +1; NS orientation and E position = 0).

*5.5. Weighting Factors*

The impact of the different elements included or not in the IC are more or less important depending on different factors related to them, such as: surface area, distance, wind direction, and pollutants hazardous to humans. It is, therefore, necessary to weight each element to avoid overestimating the effect of one element. The weighting is different depending on whether the element is inside or outside of the IC.

5.5.1. Weighting Inside the IC

Each element included in the IC is weighted by its proportion of the IC area. This ratio corresponds to an "area weighting factor" (*awf*) (Equation (1)):

$$awf = \frac{area\ of\ element_i\ (\mathrm{m}^2)}{area\ of\ \mathrm{IC}\ (\mathrm{m}^2)} \tag{1}$$

### 5.5.2. Weighting Outside the IC

The external roads included in the external traffic grid (composed of eight EC cells) are weighted in the same way as the road included in the IC (Equation (1)). Each EC has the same area as the IC. Therefore, for each EC, the streets belonging to it are weighted by an "area weighting factor" (*awf*).

The weighting of the "IED pollutants" for each industry *i* corresponds to the quantities of each pollutant weighted by its authorized limit value (Relative pollutants (*rp*)—see Equation (2)).

$$rp_i = \frac{Quantity\ of\ pollutant\ (\text{kg})_i}{Limit\ value\ of\ pollutant\ (\text{kg})_i} \tag{2}$$

In addition, the IED's distance from the IC allows for determining if the industry impact is higher or weaker. All industries within a 5 km radius of the IC have the relative value of each pollutant multiplied by two. This multiplication reflects the immediate impact of the industry on its environment. This correction is applied to the global effect from the IED industry (carcinogen effect plus quantity effect—see Section 6.1.5).

## 6. Index Construction

The index is the sum of five terms. Each term corresponds to a parameter calculated with the constituent variables presented before. The method for calculating each parameter is presented in Section 6.1. After their calculation, the parameters are normalized between 0 to 1 to allow for the AQOI determination.

### 6.1. Determination of Parameter Values before Normalization

#### 6.1.1. Topography Parameter (Tp)

The variables used are Waterway presence (*W*) and Landform (*L*). The waterway value will be weighted by the area weighting factor (*awf*) and the landform-flank value will be determined by the wind (Equation (3)):

$$Tp = \left( W \times \frac{awf}{3} \right) + L \tag{3}$$

where $W = 0$ or $1$; $L = [-1,1]$.

To not minimize the impact of the river traffic, the maximum possible waterway area in the IC is limited to one-third of the IC area.

#### 6.1.2. Building Parameter (Bp)

The building parameter is calculated through four steps:

1.　The screen effect is calculated for each building *i* from the following variables: Height ($H_i$) and Matrix Position ($MP_i$) (value following the situation: position and wind direction). This effect is calculated as a function of each frequency of the 12 *j* wind directions ($f_j$) (Equation (4)):

$$Bp_{1,i} = \sum_{j=1}^{j=12} H_{i,j} \times MP_{i,j} \times f_j \tag{4}$$

where $H = 1, 2, 3$ or $4$; $MP = -1, -0.5, 0.5$ or $1$; $f_j = [0,1]$.

2.　The heating effect is calculated for each building *i* as the Heating Area (*HA*) multiplied by a weighting factor attributed according to the fuel used (*fuel*) (this factor is calculated based on the distribution of the average energy consumption for the heating system at a regional level, e.g., Wallonia) (Equation (5)):

$$Bp_{2,i} = fuel \times HA_i \tag{5}$$

where *HA* = data extracted from GIS; *fuel* = [0,1].

3.     For each building $i$, a value is calculated including all the variables linked to the building: $Bp_1$, $Bp_2$, Gallery ($G$), and Green wall ($GW$). Only $Bp_2$ is weighting by the IC area multiplied by the housing consumption by the square meter (39,083 in Wallonia); the other variables are weighted by the area weighting factor ($awf$) of the building $i$. The value of the building parameter ($Bp$) corresponds to the sum of each contribution of the different building present in the IC (Equation (6)):

$$Bp = \sum_{i=1}^{i=n}(Bp_{1,i} + G_i - GW_i) \times awf_i + \left(\frac{Bp_{2,i}}{39,083}\right) \tag{6}$$

where $G$ = 1 or 0; $GW$ = 0 or 1.

### 6.1.3. Road Parameter (Rp)

The road parameter is calculated in three steps:

1.     The canyon effect is calculated for each street $i$ from the following variables: Canyon Type ($CT_i$) (following the ratio H/W), Canyon Length ($CL_i$) (following the ratio L/H), Matrix Orientation ($MO_i$) (value following the situation: orientation and wind direction). This effect is calculated as a function of each frequency of the 12 $j$ wind directions ($f_j$) (Equations (7) and (8)):

$$x_{i,j} = MO_i \times f_j \tag{7}$$

$$y_{i,j} = x_{i,j} + CT_i + CL_i \tag{8}$$

where $MO$ = 0, 0.25, 0.75, or 1; $f_j$ = [0,1] | $CT$ = 0, 1, 2, or 3; $CL$ = 0, 0.5, 1, or 2.

2.     Traffic is added to Equation (8) through the variables Classes ($C$) and Traffic Stop ($TS$) (Equation (9)):

$$z_{i,j} = y_{i,j} + C_i + TS_i \tag{9}$$

where $C$ = 0, 0.3, 0.5, 0.8 or 1; $TS$ = 0 or 1.

3.     The parameter value ($Rp$) is the sum of the $z_i$ value multiplied by the area weighting factor ($awf$) of street $i$ (Equation (10)):

$$Rp = \sum_{i=1}^{i=n} z_{i,j} \times awf_i. \tag{10}$$

### 6.1.4. Vegetation Parameter (Vp)

The vegetation parameter ($Vp$) is calculated in three steps:

1.     For each element $i$, a value ($Vp_{1,i}$) is calculated, taking into account the type of vegetation ($T$), the presence or not of deciduous leaves ($DL$), and the type of canopy ($C$) (Equation (11)):

$$Vp_{1,i} = (T_i + C_i) \times DL_i, \tag{11}$$

where $T$ = $-1$, 0 or 1; $C$ = 0 or 1; $DL$ = 0.5 or 1.

2.     The screen effect is calculated, for each element $i$, following Equation (12). This equation considers the position ($P$), the linear orientation ($LO$) of the vegetation and the screen factor ($Sf$) combining height, the deciduous leaves and disposition of the vegetation. All is multiplied by each frequency of the 12 $j$ wind directions ($f_j$):

$$Vp_{2,i} = \sum_{j=1}^{j=12}(LO_i \times P_i \times Sf_i \times f_j) \tag{12}$$

where $LO$ = 0, 0.5, or 1; $P$ = $-1$, $-0.5$, 0.5, or 1; $Sf$ = 1, 1.5, 2, 3, or 4; $f_j$ = [0,1].

3.     The results of Equations (11) and (12) are summed and multiplied by the weighting factor of each vegetation $i$. The vegetation parameter $Vp$ is the sum of the final contribution of vegetation $i$ (Equation (13)):

$$Vp = \sum_{i=1}^{i=n} [(Vp_{1,i} + Vp_{2,i}) \times awf_i] \tag{13}$$

6.1.5. External Sources Parameter (ESp)

The external sources parameter (*ESp*) is calculated in three steps:

1. External roads' contribution corresponds to the first part of the calculation. For each external road *i*, a value ($ESp_{1,i}$) is calculated, taking into account the road orientation (*RO*) and the road position (*RP*), all weighting by *awf*. The contribution of external roads is calculated as a function of each frequency of the 12 *j* wind directions ($f_j$) (Equation (14)):

$$ESp_1 = \sum_{i=1}^{i=n} \sum_{j=1}^{j=12} (RO_i \times RP_i \times awf_i \times f_j) \tag{14}$$

   where *RO* = 0 or 1; *RP* = 0 or 1.

2. IED industries' contribution corresponds to the second part of the calculation. For each IED industry *i*, a value ($ESp_{2,i}$) is calculated, taking into account the carcinogen effect (*C*), the relative quantity of the pollutant (*rp*—see Section 5.3.2), all weighting by the distance (*D*). The contribution of IED industries is calculated as a function of each frequency of the 12 *j* wind directions ($f_j$) (Equation (15)):

$$ESp_2 = \sum_{i=1}^{i=n} \sum_{j=1}^{j=12} \frac{(C_i + rp_i) \times f_j}{D_i} \times a \tag{15}$$

   where *C* = 0, 2.5, 5, 7.5, or 10; *rp* = [0, ... ]; *D* = [0,15]; *a* = 1 if *D* > 5 else 2; $f_j$ = [0,1].

3. The results of Equations (14) and (15) are summed to give the final value of *ESp* (Equation (16)):

$$ESp = ESp_1 + ESp_2 \tag{16}$$

*6.2. Calculation of Index AQOI*

Each parameter gives a value specific to the case study (calculated value). This value is normalized following the worst and the best case considered for the relevant parameter (the values are summarized in Table S3). For example, the worst case (maximum value) for the topography parameter is when the study site has a waterway and is located in a valley bottom. The best case (minimum value) is the absence of a waterway and hilltop.

Equation (17) is applied to each parameter's value and provides a final value between 0 to 1 (a higher value means that the impact on air quality is more negative):

$$Normalization : \frac{(parameter's\ value_{Min} - parameter's\ value_{calc})}{(parameter's\ value_{Min} - parameter's\ value_{Max})} \tag{17}$$

These values, normalized, contain information from variables associated with either sources or dispersion. Information has been adjusted following the weighting factor associated with it (see Section 6.1). The AQOI value is calculated as a sum of these five values, normalized. The index value is between 0 and 5, with 0 for situations with good air quality and 5 for situations with bad air quality.

## 7. Preliminary Tests

*7.1. Experimental Sites*

The AQOI index was tested at three sites: Jemeppe/Meuse (Figure 5b), Engis (Figure 5c), and Boncelles (Figure 5d). The selection of Jemeppe/Meuse and Engis was based on the presence of a fixed network air quality station. The station of Jemeppe/Meuse belongs to the urban category and Engis to the industrial category. Boncelles represents a rural site.

Seraing (Figure 5a) is the site used for the AQOI development (experimental site). The climatic conditions for all sites are the same as for Seraing. Figure S3 shows the positions of the different sites (orthophoto map and hill shade).

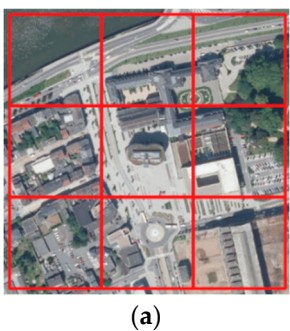 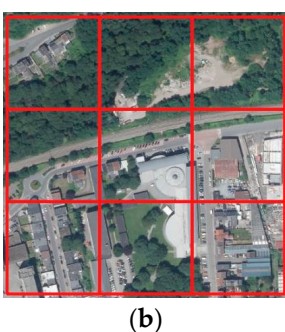 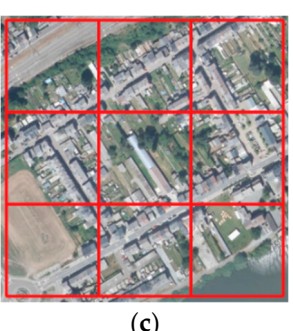 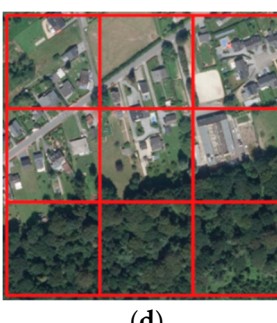

| (**a**) | (**b**) | (**c**) | (**d**) |

**Figure 5.** (**a**) Seraing; (**b**) Jemeppe/Meuse; (**c**) Engis; (**d**) Boncelles (Service Public de Wallonie, Orthophotos 2018—Service de visualisation REST. Visualization with QGIS V3.16.7).

The first site is located in Jemeppe/Meuse (50°36′ N; 5°28′ E) and is administratively part of the municipality of Seraing. The site is 0.6 km from the administration office of Seraing, across the water. The second site is located in Boncelles (50°34′31″ N; 5°32′07″ E) and is also part of the municipality of Seraing. This site is further from the administration office, 6.1 km from Seraing, and is not situated on the valley floor. The last site is located in Engis (50°35′ N; 5°25′ E) and belongs to the municipality of Huy. The site is 8.8 km from Seraing, across the water.

### 7.2. AQOI Values

The AQOI values for the four sites are calculated according to Section 6.2. The normalized parameters and AQOI values are presented in Table S4.

Following the AQOI values, Engis is the most polluted site and Boncelles the least polluted. Knowing that Engis is the most industrialized site of the four selected sites and that Boncelles is the most rural site, this observation is coherent.

To validate this first finding, a comparison with the BelAQI results was carried out. To perform this comparison, a factor called "*Impact*" was calculated using Equation (18). It is possible to calculate this "Impact" factor for each year as shown in Table S5 for Engis (BelAQI calculated based on the air quality station from Engis).

$$Impact = \sum_{BelAQI=5}^{10} RR_{BelAQI} \times number\ of\ days \qquad (18)$$

The *RR* value (increase in daily mortality from 0 μg/m³ concentration (%)) in Equation (18) represents the relative risk linked to the concentration of the pollutant [18,62].

The Impact Factor method is used to rank stations according to their impact on health. This method is only applicable when the index can be calculated.

Except for Engis, there are no official BelAQI stations in Boncelles, Jemeppe, and Seraing. We decided to select the Habay station, a rural station, for the Boncelles comparison, and the Val Benoit station for the Jemeppe and Seraing comparisons, seeing that it is located near the two studied sites. Figure 6 shows the distribution of the impact value from 2013 to 2019. The values at Engis were higher than those at Val Benoit and Habay and the one at Val Benoit was higher than that at Habay. This shows that the classification of the four sites made by the AQOI follows the trend shown by the impact value.

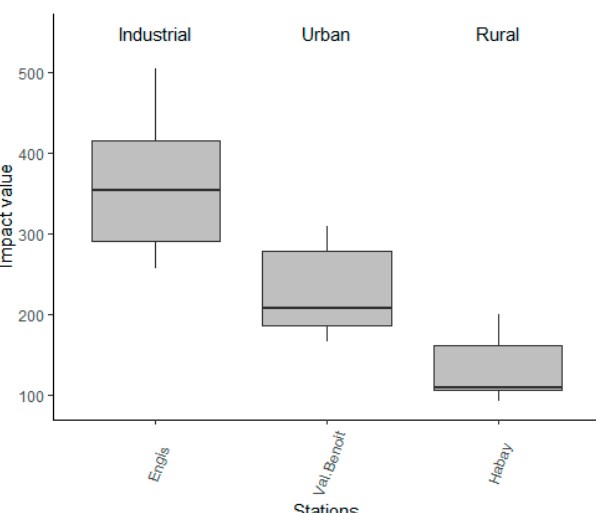

**Figure 6.** Distribution of the impact values from 2013 to 2019 of Engis, Val Benoit, and Habay air quality stations.

It is important to note that, in most cases, ozone and $PM_{2.5}$ are responsible for the poor index of rural sites. $NO_2$ is also responsible for the poor index of urban stations, such as Val Benoit. $PM_{10}$ becomes a significant cause in the case of industrial plants, such as Engis (Figure S4).

## 8. Conclusions and Future Perspectives

For the first time, assessing the air quality based on proxy-data is explored. Air quality is often obtained by measurement, modeling, or an indicator based on chemical concentrations. The index developed in this work is the first of its kind. It uses variables considered in the Land Use Regression model in addition to new variables influencing air quality. Some of them are also used in modeling, as in Computational Fluid Dynamics; however, the values are expressed differently and need not be as accurate as those required by the CFD model.

This paper describes the first phase in AQOI development: identification of the variables and their different interactions. Based on the bibliography, values have been attributed to different ordinal variables or to situations encountered in the pooling of nominal variables. The current values of the AQOI index give a coherent classification of the air quality for the four studied sites. The next steps of this research will be to conduct a sensitivity analysis of the parameters and their constituent variables to improve the AQOI and validate it with the BelAQI, which is the official air quality index in Belgium (based on $NO_2$, $O_3$, $PM_{10}$ and $PM_{2.5}$ concentrations). These steps do not exclude the possibility of adapting the values attributed to the different variables.

The size of the IC cell in which the AQOI is calculated, and which is currently 300 m × 300 m, was selected for its function in the Land Use Regression model. A study with a lower spatial resolution (a minimum value of 100 m × 100 m) will be conducted to determine the limits within which the AQOI value makes sense.

AQOI has been developed in Belgium with Belgian and European databases. It is planned to adapt it also for French sites. It will be necessary to search comparable databases (e.g., equivalence of PICC).

The information data necessary to feed the AQOI can be introduced, viewed, and selected by a GIS. Currently, it takes time to encode the IC in the Microsoft ACCESS tool. It would be interesting to develop, as soon as possible, a way to automatically extract the data using a GIS. This would save time and, in return, the GIS could show the AQOI through a heat map after running the AQOI algorithms through routines.

This paper has demonstrated that it is possible to classify different sites regarding their air quality status as a function of proxy data and open databases. The applications of this development are promising, and the next steps of the work are already underway.

**Supplementary Materials:** The following supporting information can be downloaded at: https://www.mdpi.com/article/10.3390/atmos13091470/s1, Table S1: The scales of air quality indices: USA, EU, and Belgium (BelAQI); Table S2: Descriptor for "Assignment" variable used in Belgium; Table S3: Minimum and maximum parameter values; Table S4: Parameters and AQOI values for each site; Table S5: Impact factor calculated for Engis following Equation (18). Values of BelAQI index; RR value represents the increase in daily mortality from 0 $\mu g/m^3$ concentration (%); the frequency of the index is repeated for each year; Figure S1: Flow scheme of the AQOI calculation with a GIS; Figure S2: IC divided into nine s-IC with, in its center, the building project; names of the different s-IC refer to the cardinal points of the building project; Figure S3: Situation of Seraing, Engis, Jemeppe/Meuse, and Boncelles (data Source: Service public de Wallonie—http://geoportail.wallonie.be (accessed on 30 September 2019). Visualization with QGIS V3.16.7); Figure S4: Pollutants responsible for the index according to the BelAQI calculation method. The colored areas show the presence of the pollutant responsible, and the subdivisions represent the years (e.g., pink 20 for (**a**) = occurrence of $O_3$ in 2020). (**a**) Habay air quality stations (2008 to 2020); (**b**) Val Benoit air quality station (2011 to 2020); (**c**) Engis air quality station (2008 to 2020).

**Author Contributions:** Conceptualization, methodology, software, validation, formal analysis, writing—original draft preparation, writing—review and editing, C.F.; supervision, A.-C.R. All authors have read and agreed to the published version of the manuscript.

**Funding:** This research was funded by Wallonia Region, grant number 7359.

**Institutional Review Board Statement:** Not applicable.

**Informed Consent Statement:** Not applicable.

**Data Availability Statement:** The PICC dataset is available on https://geoportail.wallonie.be/catalogue/b795de68-726c-4bdf-a62a-a42686aa5b6f.html (accessed on 6 September 2022); The E-PRTR dataset is available on https://industry.eea.europa.eu/download (accessed on 6 September 2022). The internal values related to the different calculations used to obtain the AQOI of the different sites are not published.

**Acknowledgments:** This study was part of an EcoCityTools project supported by Wallonia and GreenWin. The authors would like to thank 1Spatial for the information feeding our index, ATMPro for meteorology data, and the Scientific Institute of Public Services (ISSeP) of Wallonia for air quality data. Special thanks to the team SAM's co-workers and to Georges Lognay.

**Conflicts of Interest:** The authors declare no conflict of interest.

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
