# Peer review of "Establishing an Air Quality Index Based on Proxy Data for Urban Planning Part 1: Methodological Developments and Preliminary Tests"

_atmosphere, doi:10.3390/atmos13091470_

Round 1
Reviewer 1 Report
There are two critical issues of the article which has to be dealt with. Until than I cannot support publication of the article:
1) There is a lot of spatial variables presented in the article as well as formulas combining those spatial variables into input factors which enters the prediction model. I have to insist on description why those applied formulas are the correct ones. Why did you use those formulas and not the different ones?
2) There is no verification of the model. You calculated some values, but there is no comparison of such values with in-situ measurements. Therefore, those numbers may provide valuable insight of the air quality or be completely worthless.
Reviewer 2 Report
- Manuscript Number: Atmosphere-1835359
- Title: Establishing of an air quality index based on proxy data for urban planning. Part 1: Methodological developments
The study were to propose the AQOI (Air Quality Observed Index) without pollutant measurement. This manuscript is very interesting and fairly well described. However there seems to be some limitations, I think. I suggest that the followings be revised;
1. I fully agree with what this manuscript intends and the methodology for deriving AQOI. But, I do not agree with your opinion that air monitoring system (you described it sophisticate) and recent low-cost devices to give near-real time air quality indexes are not adapted for local air pollution and real time data are not helpful for planning. Nowadays, air pollutants concentrations and air quality index (AQI) could be provided by using of interpolation methods (GIS) with air monitoring system and low-cost devices. Of course, considering that it is high cost, it is sufficiently meaningful to present the index without measurement. However, I think that it is necessary to consider this together because air monitoring system is already installed in areas where air (atmospheric) pollution is problematic.
2. Are there anything else other than the parameters (topography, building, road, vegetation, external sources, wind direction) you suggested? For example, temperature, humidity, population by grid and so on. Especially, I think that it may be important of population exposed to air pollutants or AQI (or AQOI) by grid.
3. Can you compare the derived AQOI with the existing or conventional AQI (Air Quality Index)? In a way, I think that Figure 10 is a natural result.
4. Conclusion Section should be revised.
5. In Table 1, the names of air pollutants should be used as subscripts below.
Reviewer 3 Report
For integration of environmental considerations like air quality into green development of cities and urban planning the usual air pollutant data (NOx, PM, SO2, O3) which are measured with sophisticated monitoring stations or with low-cost devices are considered to provide an air quality index able to rank areas and projects for decision makers. It is described how such an air quality index is created as a decision support tool for urban planning. The methodological development of the so-call AQOI (Air Quality Observed Index) is provided and the results obtained for four different test cases at industrial, urban and rural locations are shown.
General comments
The AQOI is based solely on open-source data taken over one year. The introduction of this topic includes the description of some of the available air quality indices. The disadvantages of these AQI are discussed so that there is a motivation to develop a new AQI. It would be helpful to discuss, why the AQOI based on easily accessible data but with a lot of modelling, is easy to handle for non-experts.
The scientific methods and assumptions are valid and outlined mainly so that substantial conclusions are reached.
The results are sufficient to support the conclusions.
The description of experiments and analyses is complete and precise to allow their reproduction by fellow scientists.
The quality and information of the figures and tables are fine. The captions should be improved so that one can understand the figures without reading the text.
Title and abstract should mention the preliminary tests.
The overall presentation is well structured and clear. The language must be improved in detail.
The mathematical symbols, abbreviations, and units are generally correctly defined and used.
Specific Comments
It would be helpful to provide doi number for all paper and reports also if available.
The manuscript must be checked for typing errors.
Technical corrections
Do not use “et al.” in the author list
Line 717: Date of access?
Lines 708, 714, 721, 732, 734, 744, 785: References incomplete.
Line 815: Which journal?
Round 2
Reviewer 2 Report
I think it has been appropriately revised and supplemented for the inquiry made in the first review. It's somewhat different from the reviewer's (my) opinions, but I respect the authors' opinions. Somewhat more verification of this methodology is needed, and the adequacy of the research method is also needed. However, I think this paper, which proposed a new approach, has its own meaning. I hope the authors will revise and supplement the parts that are somewhat inappropriate in English expression.
Author Response
The manuscript has been submitted to the MDPI English Department for review.
Reviewer 3 Report
Lines 76-79: The CEN/TS 17660-1 is from 17/12/2021, i.e. it is available.
Author Response
Thank you, we have changed this part in the text.
The manuscript has been submitted to the MDPI English Department for review.